# Effect of Cultivable Bacteria and Fungi on the Limestone Weathering Used in Historical Buildings

Clarisse Balland-Bolou-Bi [1,*], Mandana Saheb [2], Vanessa Alphonse [1], Alexandre Livet [1], Paloma Reboah [1,2], Samir Abbad-Andaloussi [1] and Aurélie Verney-Carron [2]

1    Univ Paris-Est Creteil, Laboratoire Eau, Environnement et Systèmes Urbains (LEESU), Ecole des Ponts, Val de Marne, 94010 Créteil, France
2    Univ Paris Est Creteil and Université Paris Cité, CNRS, Laboratoire Interuniversitaire des Systèmes Atmosphériques (LISA), 94010 Créteil, France
*    Correspondence: clarisse.bolou-bi@u-pec.fr

**Abstract:** Limestone buildings in urban areas are weathered due to climatic factors, to pollution but also to biological activity. Many studies have focused on microbially-mediated precipitation of calcite but few on their influence on limestone dissolution rates. In this study, a cultivable approach for studying bacterial dissolution of limestone is proposed. The results show, for the first time, that limestone has selected a specific structure in the bacterial communities and that each bacterial class has its own metabolism inducing a different efficiency on the alteration of limestone grains. Cultivable bacterial and fungal strains in our study permit to considerably increase (by 100 to 1,000,000 times) the chemical weathering rates compared to laboratory or field experiments. Individually, the results bring information on the ability to alter limestone by dissolution. Moreover, taken together, a functional ecological adaptation of bacterial and fungal classes to the alteration of the limestone monument has been highlighted. In order to release calcium into solution, these strains slightly acidify the medium and produce low molecular mass organic acids during experiments, especially lactic and oxalic acids.

**Keywords:** cultivable strains; bioalteration; limestone



## 1. Introduction

From their establishment to the present day, historical monuments are first exposed to climatic agents (rainfall, temperature and solar radiation) leading to degradations such as loss of materials, color changes or deposits. In addition to these climatic agents, several environmental parameters can influence the alteration processes and kinetics, such as gas and particulate matter (induced by pollution) and biological activity. These phenomena take place in all environments: urban, industrial and even rural. Especially, bioalteration can induce severe damages and is generally characterized by color changes at the surface of the stone due to the pigments production [1]. In addition to the aesthetic inconvenience generated, these changes modify its sustainability.

In France, limestone is the main stone employed for the building of monument (52% according to http://monumat.brgm.fr/ (accessed on 2 January 2020)), and studying the alteration of this type of stone is a major challenge. Several studies have been carried out on the biodeterioration of limestone. Paine et al. (1932) [2] have shown that this stone serves as habitat for a wide variety of microorganisms such as bacteria with preferentially gram-negative rods and cocci depending on the environment. Most of the studies [3–6] have highlighted that heterotrophic microorganisms were the most implicated in limestone alteration. Nevertheless, sulfur-oxidizing bacteria (SOB) and nitrifying bacteria that were autotrophic were also shown to participate in the limestone weathering [2,7].

Bacteria are not the sole microorganisms involved. Fungi, microalga and lichen were detected on the surface of stone buildings [3,8,9]. The number of detectable microorganisms depends on the length of elapsed time and the season at the site. The colonization of

monuments by fungi is also influenced by the type of stone, exposure conditions (water availability) and potential nutrients (organic compounds). They can cause several forms of alteration: biofilms, black crusts, discolorations, efflorescence or pitting [10]. Microorganisms produce metabolites such as organic acids or siderophores that will induce a high stone dissolution rate by, for example, forming biofilm, permitting to keep some humidity on the surface, favoring chemical reactions (acid promoted dissolution, redox, etc.), increasing cations lixiviation [11,12]. The proliferation of fungi on limestone would occur in three phases [13,14]: a first phase with inter crystalline penetration, a second phase with intra crystalline penetration and a last phase of complete disintegration of the crystals.

Calcite ($CaCO_3$) is the main component of limestone and is very stable in the natural condition. Several studies [15–18] were performed on the biomineralisation of calcite but few studies were focused on the calcite bioweathering. In general, calcite dissolution was evaluated in seawater with model bacterial species. They have concluded that bacteria increase the dissolution rate of calcite by acidifying medium through the utilization of carbon, nitrogen and other nutrients sources and their metabolisms. However, the laboratory experiments are simplified and not representative of the real alteration processes with all the possible synergetic effects of the bacteria. Moreover, on the specific case of limestone from historical buildings which are exposed to variable environmental conditions (rain/wet cycles, thermal amplitude and relative humidity changes, seasonal variations), the action mechanisms of microorganisms on the alteration are still unknown. As the understanding of these mechanisms is a mandatory step to propose an alteration state diagnosis and adapted restoration treatments, we decided to focus on this specific subject. Moreover, most studies are carried out with 2 or 3 model microbial strains, which does not account for the versatility of microbial responses to the weathering and does not allow one to establish links between the different tested species and the involved mechanisms. The main objective of this study is thus to evaluate the contribution of all the cultivable bacterial and fungal strains that we managed to isolate, and that had a different morphotype, to the weathering of limestone grains to calculate the limestone bioweathering rates and to identify the implied mechanisms. This study will permit enriched literature data on alteration processes and permit production of a simple model to calculate and predict limestone weathering rates induced by bacteria and fungi under different conditions such as humidity that favor biological growth or pollutants contents. In addition, the relationships between the members of phyla that were capable of weathering grains will be determined using multivariate statistical analysis.

In a previous study [19], we had characterized by high-throughput pyrosequencing the bacterial composition, abundance and structure of tombs made out of limestone. We had chosen the Père-Lachaise Cemetery in Paris (France) as (1) most of the tombs (19th century) are build in limestone from the Parisian basin, (2) most of the mausoleums have not been restored, contrary to other urban historical buildings, (3) the cemetery is located in the center of Paris, in a polluted environment including an important vegetation (garden) and (4) the preservation of this historical listed site is crucial. In this study, the same limestone samples were used to select the cultivable bacterial and fungal strains.

## 2. Materials and Methods

### 2.1. Bacteria and Fungi Extraction, Purification and Identification

Limestone powder was collected from limestone surfaces with a sterile scalpel in March 2017. For extraction of bacteria and fungi from limestone collected on the selected tomb, 0.1 g of sample was mixed in 1.2 mL of sterile physiological water (NaCl 0.9 w.v$^{-1}$) in an Eppendorf® tube for 2 h. After centrifugation, 1 mL of the supernatant was sampled and diluted 10 times in sterile physiological water. Then, 100 µL of solution was redistributed on R2A agar plates (R2A agar plates double wrapping; VWR chemicals) for bacteria and PDA plates (Potato Dextrose Agar, ROTH) for fungi and incubated at 30 °C for 2 days in aerobic conditions in order to isolate pure cultivable bacterial strains. Several bacterial and fungal colonies were isolated and purified according to their morphotypes. Selected

colonies were stored at $-80\,^{\circ}$C until their utilization and identification in a specific medium (broth medium diluted 10 times for bacteria and malt extract for fungi supplemented with 20% (w.v$^{-1}$) glycerol).

Genus identification of bacteria was performed from the 16S rRNA genes amplification and their sequencing. Fast DNA spin kit (MP Bio®) was used to extract DNA from each strain according to manufacturer's instructions. Agarose gel electrophoresis (1%) was used to concentrate and purify the extracted DNA (2 µL) and the DNA amount was checked by microspectrophotometry (absorbance at 260 nm on a NanoDrop 1000, Thermo Fisher Scientific).

PCR were performed on a 1 µL cell extract and universal primers. The total reaction volume contained $1\times$ Taq PCR Master Mix (Qiagen®), 0.1 µM of each primer (27 F 5'-AGAGTTTGA TCATGGCTCAG-3' and 1100 R 5'-TTGCGCTCGTTGCGGGACT-3') for a total quantity of 50 µL. Before sequencing by the MWG Biotech Company (Courtaboeuf, France), the PCR products were purified and concentrated using mini-columns (High PureTM PCR product Purification Kit, Roche diagnostic).

Fungi identification was performed from the 16S rRNA genes amplification and their sequencing. A fast DNA spin kit (MP Bio®) was used to extract DNA from each strain according to manufacturer's instructions. Agarose gel electrophoresis (1%) was used to concentrate and purify the DNA extracted (2 µL) and the DNA amount was checked by microspectrophotometry (absorbance at 260 nm on a NanoDrop 1000, Thermo Fisher Scientific). PCR were performed on 1 µL DNA extract and universal primers. The total reaction volume contained $1\times$ Taq PCR Master Mix (Qiagen®), 0.1 µM of each primer (ITS4 5'-TCCTCCGCTTATTGATATGC-3' and ITS5 5'-GGAAGTAAAAGTCGTAACAAGG-3') for a total quantity of 15 µL. Before sequencing by MWG Biotech Company (Courtaboeuf, France), the PCR products were purified and concentrated using mini-columns (High PureTM PCR product Purification Kit, Roche diagnostic).

A blast program was used to compare the sequences with those of the GenBank databases (www.nbci.mlm.nih.gov.blast (accessed on 31 March 2015)).

### 2.2. Materials

The Saint-Maximin rock fine limestone, widely used for the replacement of stones from monuments in Paris, was used to perform stone bioweathering. This limestone from the Lutetian period (45 My) is relatively homogeneous from a chemical and physical point of view [20–23]. It is mainly composed of quartz (5%) and calcite (95%) [24]. For these experiments, the limestone was crushed and sieved (5 to 100 µm), then rinsed and sonicated with distilled deionized water to remove fine particles.

### 2.3. Experimental Set up of Limestone Biodissolution

Limestone biodissolution experiments with bacterial strains were performed in cell culture flasks (Corning™) at a temperature of 24 °C. Prior to inoculation in each flask, bacterial strains were cultivated in a 2 mL LB liquid medium (Luria and Bertani broth) during 24 h in aerobic. The bacterial cells were recovered and centrifuged to remove LB medium. Then, 2 mL of sterilized distilled water was added to rinse the cells (3 times). Finally, the bacterial cells were resuspended in 1 mL of the medium in order to reach $10^8$ cells/mL. For the fungal experiment, fungi were transferred to Melin Norkans agar. After 2 weeks of growth, fungal plugs of 5 mm in diameter were cut from the edges of the colony and used to inoculate each flask.

Each flask contained 25 mL of medium homemade based on the rainfall composition and 100 mg of limestone grains. The composition of this medium is in ppm: $K^+$ 9.8, $Na^+$ 2.91, $Mg^{2+}$ 1.79, $Al^{3+}$ 0.09, $NH_4^+$ 1.58, $PO_4^{3-}$ 0.32, $SO_4^{2-}$ 4.22, $Cl^-$ 16.59, $NO_3^-$ 3.31. Glucose (1 g.L$^{-1}$), the source of carbon and energy, was added. Limestone biodissolution experiments were conducted with an actively growing bacterial culture freshly rinsed as described below. An abiotic control was made with limestone and medium without bacteria. The duration of the experiments was 7 days. Each experiment was performed

3 times. After 7 days, solutions were sampled and centrifuged. These supernatants were filtered at 0.2 μm (filter PTFE, VWR) before storing at 4 °C.

### 2.4. Solution Analyses

During limestone dissolution experiments, pH was measured in solution (apparatus 744 pH meter metrohm®). Ca and Si released in solution was measured using an ICP-OES (Spectroblue®). The detection limits for these elements were 0.005 ppm.

Biological activities in these experiments were followed using the measurement of consumption of glucose and the production of low molecular mass organic acids (LMMOAs) in solution. The glucose concentration was measured in 10 μL aliquot of the filtrate after addition of 1 mL of GOD-PAP (enzymatic kit, BioLabo). After 20 min, absorbance was measured at 520 nm (Genesys 10 UV scanning device®). The detection limit for glucose was 0.1 ppm. According to the manufacturer's instructions, calibration curves were made to calculate the concentration of glucose (data not shown).

The LMMOAs quantification was done according to Van Hees et al., 2005 [25], using High Pressure Liquid Chromatography (HPLC). Briefly, samples were run on a C18 stationary phase (AQUASIL C18, 5 μm, 250 × 4.6 mm) at 30 °C using a mobile phase of 1% ACN/99% 0.05 M $KH_2PO_4$, pH 2.8 at a flow rate of 1.00 mL.min$^{-1}$. The LMMOAs tested were oxalic, malic, maleic, malonic, succinic, fumaric and citric acid for a wavelength of 210 nm (UV detector).

### 2.5. Solid Analyses

After the experiment, the morphology and the elemental composition of the samples were investigated using a tabletop SEM TM3030 Hitachi that is a low-vacuum scanning electron microscope equipped with energy dispersive spectrometer Quantax 70 EDS Bruker. For the measurements, an accelerating voltage of 15 kV and an accumulation time of 60 s in charge-up reduction mode were used.

Limestone was also acid-digested to quantify the proportion of Ca in the sample. This method consisted in dissolving 50 mg of limestone powder in a mixture of concentrated acids ($HNO_3$-HCl, 3-3 mL) for 48 h at 80 °C. The mixture was evaporated to dryness. All obtained residues were dissolved in 10 mL $HNO_3$ 5% before chemical analyses by ICP-OES.

### 2.6. Statistical Analysis

The normality of data was checked by using the Shapiro–Wilk normality test. Then, one-way ANOVA and the Tukey test ($p = 0.05$) for multiple comparisons were used. The statistical analyses were run using XLSTAT®.

## 3. Results

### 3.1. Identification of Bacterial and Fungal Strains

Forty-five bacterial isolates were molecularly identified based on 16S rRNA gene fragments (approximately 600 bp) from limestone samples collected from the stele. These isolates were phylogenetically affiliated to 17 different genera. Twenty-six species belonging to 17 genera were detected (Table 1). These species presented a similarity of more than 97%. Many isolated strains have been found to be of the same species whereas the morphotype of each isolate seemed different, in particular, *Pantoae agglomerans* (9 colonies identified), *Microbacterium* sp. (4 colonies identified) and *Streptomyces* sp. (3 colonies identified). Thus, 27 OTUs belong to three phyla. Proteobacteria represent the dominant phylum with 18 OTUs identified; actinobacteria and bacteroidetes are also represented with 6 and 2 OTUs, respectively. They were commonly found in soil and water and implied in organic matter decomposition [26].

**Table 1.** Identifying characteristics of the bacterial and fungal strains used in this study and the number of acquisitions obtained in the GenBank databases (http://www.nbci.mlm.nih.gov.blast (accessed on 31 March 2015)).

| PHYLUM | CLASS | ORDER | FAMILY | Closest Match According to 16S rRNA Gene Sequence | Number of Colony Identified | Accession Number | % Similarity |
|---|---|---|---|---|---|---|---|
| Proteobacteria | Betaproteobacteria | Burkholderiales | Oxalobacteriaceae | *Burkholderia* | 1 | AY839565.1 | 97 |
| | | | | *Massilia* sp. | 2 | FR865957.1 | 99 |
| | Bacilli | | Bacillaceae | *Bacillus simplex* | 2 | KJ586283.1 | 99 |
| | | | | *Bacillus licheniformis* | 1 | MF581456.1 | 98 |
| | | | | *Bacillus muralis* | 2 | KM036074.1 | 96 |
| | | | | *Bacillus* sp. | 1 | HM566651.1 | 99 |
| | | | Paenibacilliaiceae | *Paenibacillus* sp. | 2 | KY446062.1 | 98 |
| | | | | *Microbacterium* sp. 1 | 1 | KR085857.1 | 97 |
| | | | | *Microbacterium* sp. 2 | 4 | KM035942.1 | 99 |
| | | | | *Stenotrophomonas maltophilia* | 1 | LN867305.1 | 99 |
| | | | | *Stenotrophomonas* sp.1 | 1 | KC464789.1 | 99 |
| | | | | *Stenotrophomonas* sp.2 | 2 | KX588618.1 | 99 |
| | Gammaproteobacteria | Pseudomonales | Pseudomonadaceae | *Pseudomonas gessardii* | 1 | KT184489.1 | 99 |
| | | | | *Pseudomonas* sp. | 1 | KR006341.1 | 98 |
| | | Enterobacteriales | Enterobacteriaceae | *Pantoea agglomerans* | 9 | KT075163.1 | 99 |
| | | | | *Pantoea vagans* | 1 | KY127412.1 | 98 |
| | | | | *Enterobacter* sp. | 1 | KR189819.1 | 98 |
| | | | | *Erwinia billingiae* | 1 | HQ256807.1 | 97 |
| Actinobacteria | Actinobacteria | Actinomycetales | Brevibacteriaceae | *Brevibacterium frigoritolerans* | 1 | HQ202870.1 | 99 |
| | | Micrococcales | Micrococcaceae | *Arthrobacter* sp. 1 | 1 | KC019208.1 | 99 |
| | | | | *Arthrobacter* sp. 2 | 2 | HQ202815.1 | 99 |
| | | Microbacteriales | Microbacteriaceae | *Clavibacter michiganensis* | 1 | NR_133729.1 | 98 |
| | | | | *Curtobacterium* sp. | 1 | KR906476.1 | 98 |
| | | Streptomycetales | Streptomycetaceae | *Streptomyces* sp. | 3 | GU211901.1 | 98 |
| Bacteroidetes | Sphingobacteria | Sphingobacteriales | Sphingobacteriaceae | *Pedobacter* sp. | 1 | HF548384.1 | 98 |
| | | | | *Rhodoccocus* sp. | 1 | JF923558.1 | 97 |
| PHYLUM | CLASS | ORDER | FAMILY | Closest Match According to ITS Sequence | Number of Colony Identified | Accesion Number | % Similarity |
| Ascomycota | Sordariomycetes | Hypocreales | Cordycipitaceae | *Beauveria bassiana* | 1 | NR_111594.1 | 98 |
| | | | Stachybotryaceaea | *Stachybotrys chartarum* | 1 | NR_145083.1 | 99 |
| | | | Hypocreacea | *Trichoderma* sp. | 2 | NR_077207.1 | 100 |
| | Eurotiomycetes | Eurotiales | Aspergillaceae | *Penicillium speluncae* | 1 | NR_172035.1 | 94 |
| | | | | *Penicillium glandicola* | 4 | NR_119395.1 | 100 |
| | | | | *Penicillium citrinum* | 1 | NR_121224.1 | 99 |
| | | | | *Aspergillus tubingensis* | 1 | NR_131293.1 | 99 |
| | Dothideomycetes | Pleosporales | Cucurbitariacea | *Neocucurbitaria irregularis* | 1 | NR_160337.1 | 97 |
| | | Cladosporiales | Cladosporiaceae | *Cladosporium* sp. | 3 | NR_119730.1 | 99 |
| | | Dothideales | Saccotheciaceae | *Aureobasidium melanogenum* | 1 | NR_159598.1 | 98 |
| | | | | *Aureobasidium* sp. | 1 | NR_159598.1 | 71 |
| Mucoromycota | Mucoromycetes | Mucorales | Mucoraceae | *Mucor circinelloides* | 1 | NR_126116.1 | 64 |

Nineteen fungi were isolated. However, after sequencing, only 12 fungi were identified (Table 1). Unfortunately, several strains corresponded to the same species, although their morphotype appeared to be different. *Penicillium glandicola* is the most represented species with 5 colonies identified. It is followed by *Cladosporium* sp. and *Trichoderma* sp. with 3 and 2 colonies identified, respectively. Among these 12 species, 9 genera were identified.

*3.2. Bacterial and Fungal Control of Limestone Dissolution*

In general, bacterial and fungal strains induced the leaching of Ca from limestone (Figure 1A,B). Among the 26 different bacterial strains tested, only 16 strains are presented because the other 10 strains were not able to survive during the precultivation step in the enriched medium. The quantity of Ca released into solution (expressed in percentage) comprised between 0.4% to 2.2% in 7 days. The strains *Clavibacter michiganensis* sp., *En-*

*terobacter* sp., Stenotrophomonas maltophilia, Microbacterium phyllosphaere, *Rhodococcus* sp., *Streptomyces* sp., *Bacillus* sp. and Bacillus simplex are in the same range as the abiotic control (0.40 ± 0.15%) and have no significant effect on the release of Ca from limestone in solution in comparison to the other strains. Burkholderia, *Massilia* sp., Bacillus licheniformis, *Microbacteriums p.*, *Stenotrophonas* sp., *Pseudomonas gessardii*, *Pantoae agglomerans* and *Pedobacter* sp. are more efficient, with a Ca release between 2 to 5 times more important than abiotic control. Concerning Si, the percentage of leached Si reached a value range of approximatively 0 to 0.013% in 7 days (Supplementary Data S1). All the strains except *Stenotrophomonas maltophilia* and *Pseudomonas gessardii* are in the same range as abiotic control (0.005%) and have no effect on the release of Si from quartz in solution (data not shown). Correlations between Si released and the other parameters have not been found.

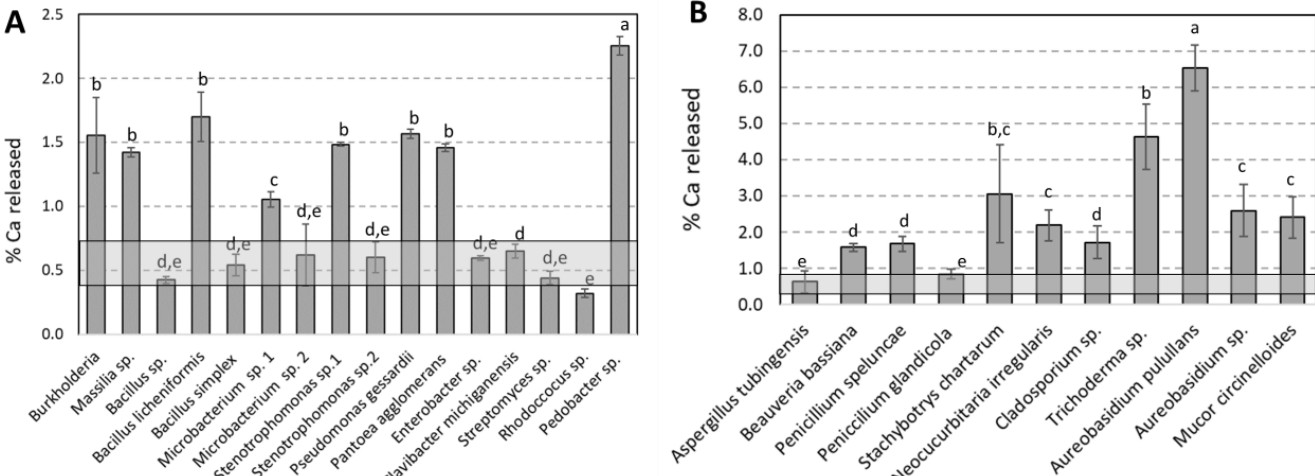

**Figure 1.** Percentage of Ca released for (**A**) each bacterial and (**B**) fungal species tested and abiotic control. Each bar corresponds to the mean and standard deviation calculated from three replicates. The symbols a, b, c, d and e correspond to the different groups of significance obtained by ANOVA 1 factor and by the Tukey test ($p$ = 0.05) on three replicates. The grey transparent band corresponds to the mean and standard deviation of abiotic control.

Concerning experiments performed with fungal strains, the quantity of Ca released into solution (expressed in percentage) comprised between 0.6% to 6.5% in 7 days, that is, 3 times more than bacterial strains. The strains *Aspergillus tubingensis* and *Penicillium glandicola* are in the same range as the abiotic control (0.60 ± 0.2%) and have no significant effect on the release of Ca from limestone in solution in comparison to the other strains. *Beauveria bassinia*, *Penicillium speluncae*, *Stachybotris chartarum*, *Neocucurbitaria irregularis*, *Cladosporium* sp., *Thricoderma* sp., Aureobasidium pullulans, *Aureobasidium* sp. and *Mucor circinelloides* are more efficient with a Ca release between 2 to 10 times more important than abiotic control. *Aureobasidium pullulans* was the most efficient fungal strain. Concerning Si, the percentage of leached Si reached a value range of approximatively 0 to 0.42% in 7 days (Supplementary Data S1). All the strains are in the same range as abiotic control (0.005%) and have no effect on the release of Si from quartz in solution (Supplementary Data S1) except *Aspergillus tubingensis* and *Penicillium glandicola*. These 2 strains were, however, the least effective in releasing, indicating a preferential leaching of Si. No correlations between Si released and the other parameters have been found.

In addition, a positive correlation between the percentage of Ca released and of glucose consumed has been highlighted (Figure 2A) for both bacteria and fungi indicating a biological control of limestone weathering.

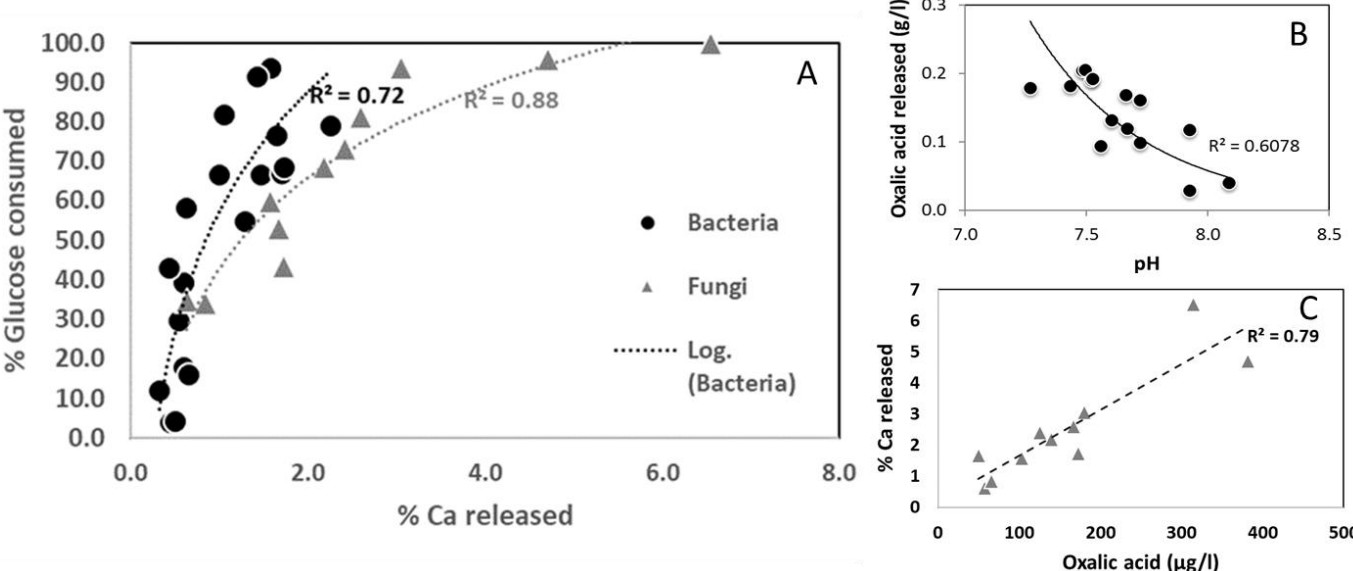

**Figure 2.** Linear correlation between parameters: (**A**) Percentage of Ca released as a function of percentage of used glucose, (**B**) Concentration of oxalic acid released as a function of pH for experiments performed by bacteria and (**C**) Percentage of Ca released as a function of concentration of oxalic acid for experiments performed by fungi. Each point corresponds to value of a strain.

LMMOAs were identified and quantified after 7 days (Figure 3 and Supplementary Data S2). Only oxalic, malonic, lactic, acetic, succinic, formic, glutaric and citric acids were quantified for experiments performed with bacteria. No correlation has been found between total LMMOAs and Ca released. *Microbacteriums p.* and *Stenotrophomonas* sp. display a higher production of LMMOAs, largely influenced by lactic acid (3098 and 2317 µg.L$^{-1}$, respectively) in comparison to the other strains. This high production of lactic acid did not seem to stimulate calcium leaching from limestone by these two bacterial strains (Figure 1). Whatever the bacterial strain, oxalic acid represents the main produced organic acid, followed by lactic, formic, succinic, citric, malonic, acetic and glutaric acids. Oxalic acid is the sole LMMOA quantified in experiments performed by *Massilia* sp., *Bacillus simplex*, *Pantoae agglomerans*, *Clavibacter michiganensis*, *Streptomyces* sp., *Rhodococcus* sp. and *Pedobacter* sp. There is a correlation between released oxalic acid and the acidification of the medium (Figure 2B) for bacteria, indicating that more oxalic acid is released, more the pH is acid surpassing the buffering capacity of the medium in the presence of limestone.

Only oxalic acid was detected and quantified for experiments performed with fungi, as it represents the main produced organic acid. A very good positive correlation has been found between total LMMOAs, percentage of glucose consumed and Ca released. Stachybotrys chartarum, *Trichoderma* sp., *Aureobasidium pullulans*, *Cladosporium* sp. and *Aureobasidium* sp. display a higher production of oxalic acid (180, 381, 314, 172 and 167 µg.L$^{-1}$, respectively) in comparison to the other strains. This high production of oxalic acid combined with the use of glucose as a carbon source seem to stimulate calcium leaching from limestone by these four fungal strains. This is confirmed by the positive correlation found between these 2 parameters (Figure 2C).

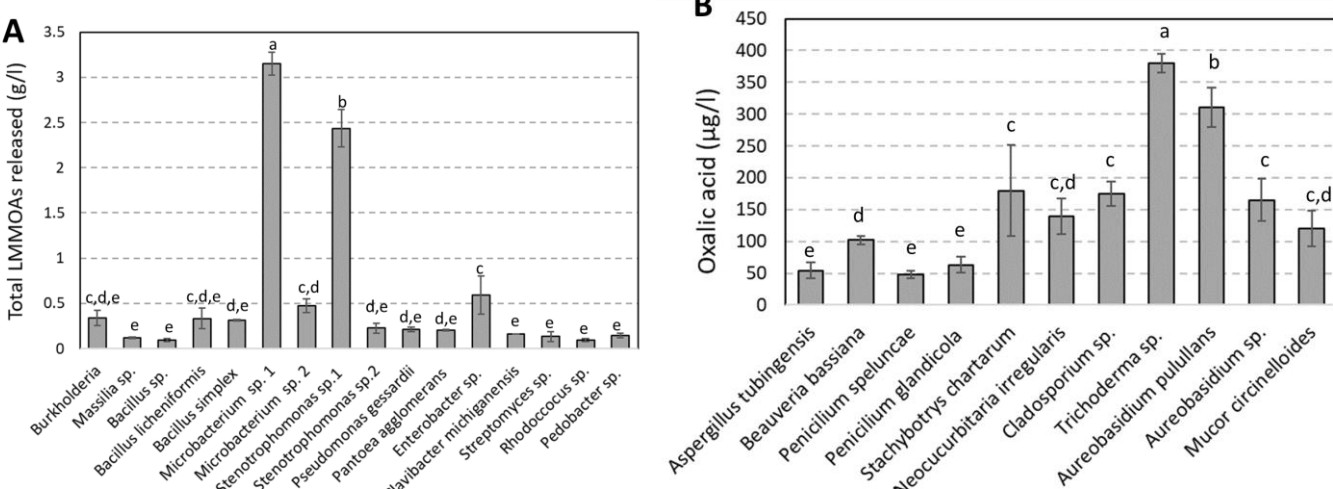

**Figure 3.** (**A**) Total LMMOAs released for each bacterial specie tested and (**B**) oxalic acid released for fungal strains. Each bar corresponds to the mean and standard deviation calculated from three replicates. The symbols a, b and c correspond to the different groups of significance obtained by ANOVA 1 factor and by the Tukey test ($p = 0.05$) on three replicates.

### 3.3. Biocolonisation Assessment

A set of limestone samples representative of each experimental condition performed by bacterial and fungal strains was observed by SEM at the end of experiments (Supplementary Data S3). The observations show the presence of organic products on the limestone surface (Supplementary Data S3B–F) compared to the initial state (Supplementary Data S3A), which is confirmed by the presence of C, S and O (EDS analyses). These micrographs also allow discriminating the crystals from the organic material, through the presence of Ca on the grain surface. According to the SEM images measured on the limestone grains after biodissolution experiments, the presence of biofilm was detected (Supplementary Data S3B,D) for experiments performed with *Rhodocossus* sp. and *Pedobacter* sp. Moreover, filaments or hyphae were also detected for experiments performed with *Stenotrophomonas* sp., *Penicillium glandicola* and *Aureobasidium* sp. Biofilms and filaments were attached with calcite crystals. They also contain C, with small amounts of Si, Ca or S. The latest element might be made of extracellular polymeric substances (EPS) secreted by *Rhodocossus* sp. and *Pedobacter* sp., during experiments, indicating a real biological activity.

## 4. Discussion

According to limestone alteration experiments, 8 cultivable bacteria (*Burkholderia*, *Massilia* sp., *Bacillus licheniformis*, *Microbacterium* sp., *Stenotrophonas* sp. 1, *Pseudomonas gessardii*, *Pantoae agglomerans* and *Pedobacter* sp.) and 9 fungi (*Beauveria bassinia*, *Penicillium speluncae*, *Stachybotrys chartarum*, *Neocucurbitaria irregularis*, *Cladosporium* sp., *Trichoderma* sp., Aureobasidium pulullans, *Aureobasidium* sp. and *Mucor circinelloides*) isolated from a tomb in the Père-Lachaise Cemetery were able to significantly dissolve limestone grains. These bacteria and fungi have been commonly identified on limestone [4,19,26–29].

These microorganisms have used glucose such as carbon source, and as a consequence, they have partially oxidized it, produce byproducts of metabolisms: LMMOAs that will lead to $CO_2$ release either during respiration or fermentation [8,30,31]. This will lead to the biodissolution of limestone by first decreasing pH and modifying the chemistry of the solution at mineral interfaces in order to leach Ca and Si as a function of microorganisms involved. A positive correlation has been highlighted between the percentage of used glucose (carbon source) and the percentage of Ca released into solution (Figure 3A) whatever the biological agent (bacteria or fungi). This correlation indicates a biological control of the limestone dissolution during experiments performed by microbial strains. Within the

same family, each microbial strain has its own metabolism inducing a different efficiency on the alteration of calcite grains. To release calcium into solution, they slightly acidify the medium and produce low molecular mass organic acids, especially lactic and oxalic acids for bacteria and oxalic acid for fungi. These weak acids increase mineral weathering through decreasing pH. Several authors [11,32] have highlighted a different action mode of LMMOAs on the rate of mineral weathering according to the minerals, the kind of ligand and its concentration. Indeed, oxalic, succinic and citric acids were more efficient in increasing the silicate mineral dissolution rate because they are composed of several carboxylic groups (polyfunctional acids), contrary to lactic or acetic acids (monofunctional acids). On the other hand, for dissolution experiments performed with limestone and LMMOAs, Huang et al. (2006) [33] have shown that the most efficient LMMOAs were the monofunctional acids such as lactic and acetic acids followed by citric, formic, oxalic and pyruvic acids. These results are contradictory with our results, as it seems that the production of lactic acid by bacterial strains was not efficient to release calcium from limestone. No correlation has been found between pH and Ca released into solution. This can be explained by the stability of calcite under earth surface conditions and its buffering effect on pH in solution [34] and on metabolic products continuously secreted by microbial strains.

The chemical weathering rates $(mol.m^{-2}.s^{-1})$ of limestone were calculated according to our results and the mean value was $3.79 \times 10^{-5}$ $mol.m^{-2}.s^{-1}$ for bacterial strains and $9.92 \times 10^{-5}$ $mol.m^{-2}.s^{-1}$ for fungal strains at T = 24 °C and pH around 7.5. After reported literature data on chemical weathering rates (Table 2), cultivable bacterial strains in our study permit to increase considerably (by 100 to 1,000,000 times) the chemical weathering rates compared to laboratory experiments performed at pH 4 to 6 [35], field experiments performed at pH 4 to 8 [36,37] and limestone bioweathering performed by lichen [38]. However, compared to limestone weathering performed by fungal mycelia [39], weathering rates are lower as fungi but in the same order than for our fungal strains. In addition of chemical action, hyphae may cause an important penetration in the stone pores, increasing the stone weathering [5].

**Table 2.** Chemical weathering rates $(mol.m^{-2}.s^{-1})$ of limestone.

| Modalities | Chemical Weathering Rates $(mol.m^{-2}.s^{-1})$ | References |
|---|---|---|
| Laboratory experiments pH 4<br>Laboratory experiments pH 5.5 | $2.25 \times 10^{-6}$<br>$9.17 \times 10^{-8}$ | Franke and Teschner-Steinhardt (1994) [35] |
| Field experiments pH 4.5<br>Field experiments pH 6<br>Field experiments pH 6.5<br>Field experiments pH 7.9 | $1.59 \times 10^{-6}$<br>$7.02 \times 10^{-9}$<br>$2.25 \times 10^{-9}$<br>$1.10 \times 10^{-10}$ | Swoboda-Colberg and Drever (1993) [37] |
| Field experiments | $1.32 \times 10^{-9}$ | Roussel and André (2013) [36] |
| Laboratory experiments Bare rock with cyanobacteria<br>Lichen-covered rock | $9.86 \times 10^{-10}$<br><br>$6.18 \times 10^{-10}$ | Fiol et al. (1996) [38] |
| Laboratory experiments by fungal mycelia | $1.54 \times 10^{-4}$ | Li et al. (2009) [39] |
| Laboratory experiments by bacterial strains:<br>*Burkholderia* sp.; *Pantoea agglomerans*; *Pseudomonas gessardii; Bacillus simplex* | $\mathbf{3.79 \times 10^{-5}}$ | **Our study** |
| Laboratory experiments by fungal strain:<br>*Aureobasidium pullulans, Trichoderma* sp.,<br>*Stachybotrys chartarum* | $\mathbf{9.92 \times 10^{-5}}$ | **Our study** |

According to these results (metabolic products) and SEM micrographs, bacterial cells and their biofilm are embedded within the calcite crystals, creating a microenvironment

that is different to the chemistry of the solution (pH, dissolved oxygen and concentrations of organic and inorganic molecules) [40–42]. These micro-reaction zones can explain the differences of efficiency between experiments performed by pure bacterial strains, as well as the metabolic pathway.

The last objective was to establish a link between taxonomic and functional characteristics of limestone-weathering bacteria and fungi. In our works, we have isolated 16 bacterial and 11 fungal strains from a tomb and tested their ability to dissolve calcium from limestone. According to factorial discriminant analyses performed on these data (limestone biodissolution and classes of the bacterial and fungal strains), each class of bacteria (Actinobacteria, Sphingobacteria, Bacilli, Betaproteobateria and Gammaproteobacteria) or fungi (Eurotiomycetes, Sordoriomycetes, Dothideomycetes and Mucoromycetes) are regrouped in clusters, indicating that the ability to weather limestone and the process implied are a common functional trait of all identified classes (Figure 4A,B). Thus, for bacteria, Sphingobacteria, Betaproteobateria and Gammaproteobacteria are the most efficient classes to weather limestone. Bacilli and Actinobacteria are the less efficient ones, despite the strong production of low molecular mass organic acids. For fungi, Dothideomycetes and Mucoromycetes are the most efficient classes to weather limestone. These results only take into account the percentage of calcium released in solution during the experiments. Calcium could also be immobilized into biofilm or Ca-oxalates. Some studies [43–45] have identified in situ, during limestone biodissolution, some alteration products linked to biological colonization including calcium oxalates, whewellite ($CaC_2O_4$, $H_2O$), weddellite ($CaC_2O_4$, $2H_2O$) and gypsum ($CaSO_4$, $2H_2O$). Calcium oxalates result from the reaction between oxalic acid produced by microorganisms and calcium released from limestone. Gypsum, on the other hand, results from the reaction of limestone with the combustion and exhaust gases ($SO_2$). However, in this study, alteration products were not detected by XRD analyses.

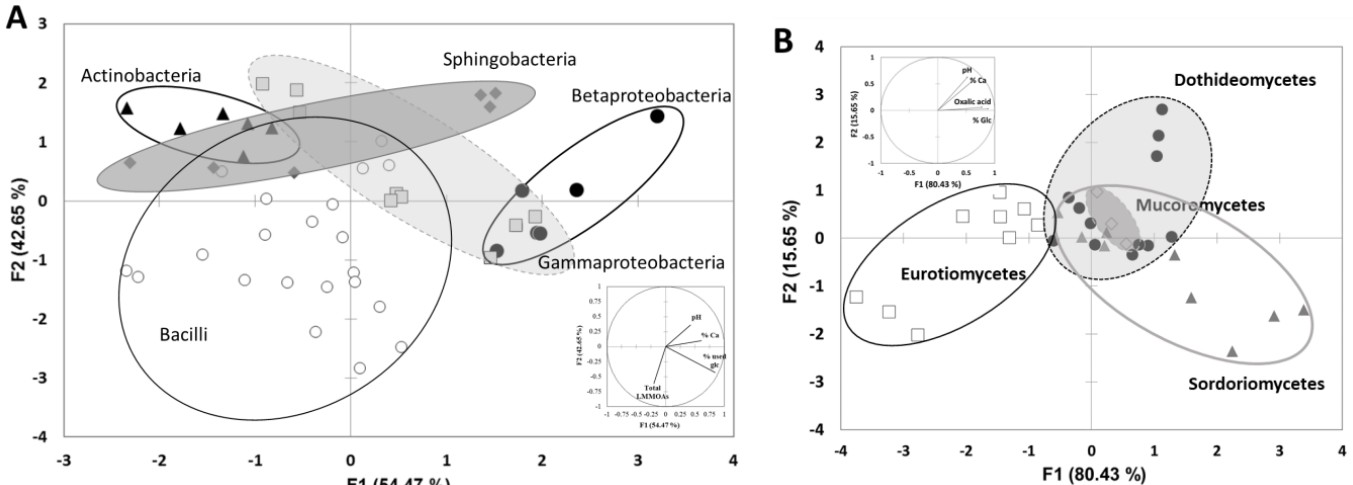

**Figure 4.** (**A**) Factorial discriminant analysis performed on data of limestone biodissolution by bacterial strains (pH, total LMMOAs released, percentage of leached Ca and percentage of glucose consumed). FDA map according to the first two factorial scores (F1 and F2) where the discriminated groups are the classes to which the bacterial strains belong (Betaproteobacteria, Gammaproteobacteria, Actinobacteria, Bacilli and Sphingobacteria). (**B**) Factorial discriminant analysis performed on data of limestone biodissolution by bacterial strains (pH, oxalic acid, percentage of leached Ca and percentage of glucose consumed). FDA map according to the first two factorial scores (F1 and F2) where the discriminated groups are the classes to which the fungal strains belong (Eurotiomycetes, Sordoriomycetes, Dothideomycetes and Mucoromycetes).

These results bring a new perspective on biocolonization and biodissolution assessment by pure bacterial and fungal strains. Individually, the results only bring information

on the ability to alter limestone, but taken together, a functional ecological adaptation of bacterial and fungal classes to the alteration of the limestone monument is highlighted. To date, few studies have attempted to functionally characterize bacteria associated with the surface of the materials [5,19,46]. Most of the time, these works are based on the literature on altered soil minerals. The cultivable approach remains essential for studying the functional capacities of bacterial and fungal isolates. Some studies [47,48] have shown a high proportion of bacterial isolates assigned to Firmicutes (45%) and β-proteobacteria (13%) associated with soil minerals biodissolution. For fungi, the most represent phylum is Asmycota and therein, the classes of Dothideomycetes [29,49]. These results are consistent with some studies that have shown a selection of specific microbes on the minerals surface [50,51]. Of course, these data remain to be confirmed by carrying out several tests with many more strains. It seemed that limestone has selected a specific structure in the microbial communities and each family has its own metabolism inducing a different efficiency on the alteration of limestone grains demonstrating a great bioreceptivity of limestone. Indeed, bioreceptivity is defined as the ability of stone materials to be colonized by living organisms [52] according to petrophysical properties and chemical composition [53]. Taking account our results, limestone is a stone that seems to have favored the colonization of living microorganisms. These results are coherent with literature data. In addition, several works [3,46,54,55] have also demonstrated that atmospheric pollutants cause a removal and a change in the biofilm microflora of different limestones, and this phenomenon accentuates this colonization even more.

## 5. Conclusions

This study demonstrates that the limestone as a medium has favored a specific structure in the bacterial and fungal communities and each class has its own metabolism inducing a different efficiency on the alteration of limestone grains. Individually, the results provide information on the ability to alter limestone; moreover, taken together, a functional ecological adaptation of bacterial classes to the alteration of the limestone monument has been highlighted. Concerning the involved mechanisms, in order to release calcium into solution, these bacteria and fungi slightly acidified the medium and produced low molecular mass organic acids during experiments, especially lactic and oxalic acids.

**Supplementary Materials:** The following supporting information can be downloaded at: https://www.mdpi.com/article/10.3390/d15050587/s1, S1: Supplementary Data S1. Percentage of calcium and silicium released into solution, Percentage of glucose consumed and pH for all the experiments; S2: Supplementary Data S2. Low molecular mass organic acids concentrations (mg/L) in limestone biodissolution by bacterial strains. "-" means below detection limits. S3: Supplementary Data S3. Scanning electron micrographs of initial (non-contacted) limestone grains (A) and after 7 days in the bioleaching medium with *Rhodococcus* sp. (B), *Stenotrophomonas* sp. (C) *Pedobacter* sp. (D), *Penicillium glandicola* (E) and *Aureobasidium* sp. (F).

**Author Contributions:** Conceptualization, C.B.-B.-B., M.S., A.V.-C.; methodology, C.B.-B.-B. and V.A.; formal analysis, C.B.-B.-B.; data acquisition, C.B.-B.-B., A.L., P.R. and V.A.; writing—original draft preparation, C.B.-B.-B., M.S. and A.V.-C.; writing—review and editing, C.B.-B.-B., M.S. and A.V.-C.; project administration, S.A.-A.; funding acquisition, S.A.-A. All authors have read and agreed to the published version of the manuscript.

**Funding:** This research was funded by French National Research Agency (ANR), funding number: MIAM project Project-ANR-19-CE22-0006.

**Informed Consent Statement:** Not applicable.

**Data Availability Statement:** All the data will be given on request.

**Conflicts of Interest:** The authors declare no conflict of interest.

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
