# Peer review of "Effect of Cultivable Bacteria and Fungi on the Limestone Weathering Used in Historical Buildings"

_diversity, doi:10.3390/d15050587_

Round 1

Reviewer 1 Report

This is an interesting and well thought out and presented paper. I only have a few minor points, mostly textual that the authors may like to address.

Lines 54 and 447 the use of elipses (...) is not common grammar  and should probably be avoided here. In line 54 etc. would fit better.

Line 95 "for" rather than "during"

For the collection/isolation of the organisms, I read that isolations were made from the tombs in the cemetry. if not please make that clearer. If that is the case what time of year, weather was the collection performed in? I wonder if that could make a difference on what species might be picked up.

Lines 236 to 239 are a little unclear on first read. I think the authors are saying that of all the specie tested only the aspergillus and penicillium showed any increase in Si in contrast to the abiotic controls. As these fungi also showed poorer release of Ca than other specie this suggests that there growth pattern preferentially leached Si rather than Ca. Is that correct. I think it can be worded more clearly if so.

Line 269to 271. This seems and obvious point to note that release of an acid makes a medium more acidic, unless the authors are suggesting there may have been some buffering capacity?

Line 324 - 325 mentions a biological control. On first reading I thought the authors were suggesting competition between organisms as per deliberate biological control. I think the authors mean that there are differing biological process occuring based on which specie is involved. If thats correct then some editing to make it clearer might be useful.

Overall a nice paper. thank you

Author Response

Dear Reviewer, we would like to thanks you for all your comments and the minor points reported. We have made all the changes asked even in the tables and figures. We have updated our references. We have also modified some structures of the sentences to make it clearer.

Best Wishes,

Clarisse Balland-Bolou-Bi and co-authors.

Reviewer 2 Report

General comments

 1. In the text, the scientific names (genus and species) must be written in italics. For example, on lines 191 and 192, it would be: ”Pantoae agglomerans (9 colonies identified), Microbacterium sp. (4 colonies identified) and Streptomyces sp. (3 colonies identified)”.

Note that the abbreviation "sp." not written in italics.

Please correct scientific names in all text and figures.

2. Concerning bacteria, in Table 1, only 26 species are cited, not 27. About Proteobacteria, in Table 1 are represented only 18 OTUs, not 19.

Please check this.

3. Please check the list of species in table 1, according with this list:

Arthrobacter sp. 1

Arthrobacter sp. 2

Bacillus licheniformis

Bacillus muralis

Bacillus simplex

Bacillus sp.

Brevibacterium frigoritolerans

Burkholderia

Clavibacter michiganensis

Curtobacterium sp.

Enterobacter sp.

Erwinia billingiae

Massilia sp.

Microbacterium sp. 1

Microbacterium sp. 2

Paenibacillus sp.

Pantoea agglomerans

Pantoea vagans

Pedobacter sp.

Pseudomonas gessardii

Pseudomonas sp.

Rhodococcus sp.

Stenotrophomonas rhizophila

Stenotrophomonas sp. 1

Stenotrophomonas sp. 2

Streptomyces sp.

4. Stenotrophomonas maltophilia, cited in lines 213 and 221, and Figures 1 and 3, is not in Table 1.

Please check this.

5. In Figure 1. Lack letters A and B in each box. In the box B, change “Stachybotrys chratrum” for “Stachybotrys chartarum

6. In Table 2, add the reference numbers. For example: Franke and Teschner -Steinhardt (1994) [36].

7. Please, in the "references" review the highlighted text

Specific changes

Line 40. “Paine et al. (1932) have..” change for “Paine et al. (1932) [2] have.. or “Paine et al. [2] have..”

Line 67. “rain/wet cycles, temperature and relative humidity changes” change for “rain/wet cycles, thermal amplitude and relative humidity changes”

Line 164. “..to Van Hees et al. 2005 using..” change for “..to Van Hees et al. 2005 [25] using..”

Figure 3. Box B, change “Peniccilium glandicola” for “Penicillium glandicola

Figure 4. Box B, change “Mucoromycota” for “Mucoromycetes”

Other changes are in the manuscript

Author Response

Dear Reviewer, we would like to thanks you for all your comments and the minor points reported. We have made all the changes asked even in the tables and figures and checked the scientific name of bacteria and fungi. We have updated our references. We have also modified some structures of the sentences to make it clearer.

Best Wishes,

Clarisse Balland-Bolou-Bi and co-authors.

Round 2

Reviewer 2 Report

Minor corrections are included in the attached document

Author Response

Dear reviewer, thanks again for your report. We have made all the changes asked. Sincerely, Clarisse alland-Bolou-Bi and co-authors
